# The Effect of Tannin-Rich Witch Hazel on Growth of Probiotic *Lactobacillus plantarum*

**DOI:** 10.3390/antibiotics11030395

**Published:** 2022-03-16

**Authors:** Reuven Rasooly, Alex C. Howard, Naomi Balaban, Bradley Hernlem, Emmanouil Apostolidis

**Affiliations:** 1U.S. Department of Agriculture, Agricultural Research Service, Albany, CA 94710, USA; bradley.hernlem@usda.gov; 2College of Engineering, University of Massachusetts Amherst, Amherst, MA 01003, USA; achoward@umass.edu; 3Department of Chemistry and Food Science, Framingham State University, Framingham, MA 01701, USA; nbalaban@framingham.edu

**Keywords:** biofilm, *Lactobacillus plantarum*, prebiotic, probiotic, witch hazel (*Hamamelis*)

## Abstract

Probiotic bacteria help maintain microbiome homeostasis and promote gut health. Maintaining the competitive advantage of the probiotics over pathogenic bacteria is a challenge, as they are part of the gut microbiome that is continuously exposed to digestive and nutritional changes and various stressors. Witch hazel that is rich in hamamelitannin (WH, whISOBAX^TM^) is an inhibitor of growth and virulence of pathogenic bacteria. To test for its effect on probiotic bacteria, WH was tested on the growth and biofilm formation of a commercially available probiotic *Lactobacillus plantarum* PS128. As these bacteria are aerotolerant, the experiments were carried out aerobically and in nutritionally inadequate/poor (nutrient broth) or adequate/rich (MRS broth) conditions. Interestingly, despite its negative effect on the growth and biofilm formation of pathogenic bacteria such as *Staphylococcus epidermidis*, WH promotes the growth of the probiotic bacteria in a nutritionally inadequate environment while maintaining their growth under a nutritionally rich environment. In the absence of WH, no significant biofilm is formed on the surfaces tested (polystyrene and alginate), but in the presence of WH, biofilm formation was significantly enhanced. These results indicate that WH may thus be used to enhance the growth and survival of probiotics.

## 1. Introduction

*Lactobacillus plantarum* (LP) are Gram-positive lactic bacteria, commonly found in fermented foodstuff and in the gastrointestinal tract. LP is used as probiotic bacteria, having medical applications due to its antioxidant, anticancer, anti-inflammatory, antiproliferative, antiobesity, and antidiabetic properties [1]. Intestinal microbiome homeostasis is essential for the maintenance of a healthy gut, where imbalances can lead to digestive, dietary, metabolic, and even mental health issues [2,3]. The gut bacterial microbiome, containing hundreds of distinct bacterial species, can be disrupted by dietary changes, stress, and antibiotics. Long-term use of broad-spectrum antibiotics can lead to the destruction of normal micro-flora while allowing antibiotic-resistant strains to overgrow. One such overgrowth that commonly results from long-term use of antibiotics is *Clostridium difficile*, which are bacteria that cause severe diarrhea, colitis, and even death. This highly contagious infection now kills one in every eleven patients over the age of 65 within a month of diagnosis [4].

Human gastrointestinal homeostasis is a delicate balance maintained by the interactions of the trillions of bacteria that colonize the gut. These “good bacteria” are collectively referred to as the microbiota [3]. Bacteria communicate with each other using quorum-sensing mechanisms to coordinate each other’s activities to better survive in any specific environment [5]. Quorum sensing, for example in staphylococci, involves the *agr* system that regulates toxin production [6] and the TRAP (Target of RNAIII Activating Protein) signaling system that protects cells from stress and from higher rates of spontaneous and adaptive *agr* mutations [7]. When the *agr* and *traP* genes are activated, the bacteria produce toxins that allow them to evade host immune response, overcome metabolic deficiencies, and overcome stressors such as the oxidative stress that is produced by the host as a response to infection [6].

The structural homologue of the protein TraP in bacilli is YhgC [8,9]. YhgC was renamed HmoB because of its sequence similarity to heme mono-oxygenase (HO) [10], which degrades heme to biliverdin, carbon monoxide, and ferrous iron, and is essential for heme and iron homeostasis. HO consumes molecular oxygen to oxidize heme, establishing an anoxic microenvironment, and is thus also necessary for adaptation to cell stress [10].

In staphylococci, the phosphorylation of TraP is up-regulated as bacterial density increases, and its phosphorylation can be downregulated by a peptide-inhibitor RIP or its non-peptide analogue Hamamelitannin. Both were shown to be extremely effective in inhibiting bacterial pathogenesis [11,12,13]. Hamamelitannin is naturally found in witch hazel (*Hamamelis*) and is especially abundant in its bark [14].

Another important mechanism for bacterial survival is the formation of a biofilm, which is essentially a cellular fortress that can be composed of a single or multiple microbial species that may communicate and coordinate their activities [15]. In a healthy gut, biofilms can promote nutrient exchange between bacteria and the host as well as increase the time the bacteria can remain in the gastrointestinal tract, allowing bacteria to attach and proliferate. This behavior of occupying that space has the added benefit of preventing pathogenic bacteria from establishing itself in the host. The host’s immune system, together with the microbiome, establishes the mucus barrier that simultaneously prevents infection and promotes probiotic proliferation [15,16,17,18].

To help maintain the healthy microbiome, the effect of a plant extract whISOBAX (WH) was tested on the growth and biofilm formation of a strain of *L. plantarum* (LP) that is commercially available as a probiotic. WH is hamamelitannin-rich witch hazel extract that was shown to inhibit cell-to-cell communication (quorum sensing) in staphylococci, limiting their ability to cause disease by suppressing their pathogenic potential [19,20,21].

## 2. Results

### 2.1. Effect of Witch Hazel on Growth and Biofilm Formation of SE

To identify a witch hazel extract that is most efficient at inhibiting the growth and virulence of pathogenic bacteria, *Staphylococcus epidermidis* (*S. epidermidis*, or *SE*) was grown with three different commercially available witch hazel extracts: WH (whISOBAX, StaphOff Biotech Inc., Hopkinton, MA, USA), CareOne (Salisbury, NC, USA), and NFH (Nutritional Fundamentals for Health, Vaudreuil-Dorion, QC, Canada).

The inhibitory effect of WH on the growth of common gut pathogens such as *Escherichia coli* and *Klebsiella pneumoniae* has been demonstrated [19]. *S. epidermidis* was used as a model for pathogenic bacteria because the effect of WH on staphylococci is well documented [19,20,21] and because *S. epidermidis* is the most frequently isolated species of the coagulase negative staphylococci from human stool and its intestinal colonizing can cause severe disease [22].

The specific witch hazel extracts tested were chosen because of their relative tannin content, with WH containing 17 mg/mL hamamelitannin [21], NFH containing 6 mg/mL hamamelitannin (data not shown), and CareOne distillate having no documented tannins.

As shown in Figure 1 and Figure 2, WH is the most effective tested extract in terms of the inhibition of growth and biofilm formation of *S. epidermidis*, with a minimal inhibitory concentration (MIC) at a dilution of 1:80. NFH was also effective but with a higher MIC (at dilution 1:20), and no inhibitory activity was found for CareOne.

WH was further tested for its effect on the growth and biofilm formation of the probiotic *L. plantarum* (LP).

### 2.2. The Effect of whISOBAX (WH) on L. plantarum (LP) Growth

To test for the effect of WH on the growth of LP, bacteria was grown aerobically in nutrient broth or in MRS, a richer culture medium. As shown in Figure 3a, WH has no significant effect on LP growth in MRS, while there is little to no growth of LP in nutrient broth without the addition of WH (Figure 3b). With the addition of WH (tested at up to a 1:80 dilution of WH), a significant (*p* < 0.01) increase in LP growth is observed. This effect is due to WH and not its control solvent since there is no change in LP growth with the addition of ethanol (tested at concentrations of up to 2.5%, in this case a 1:20 dilution of 50% ethanol). Samples were plated on MRS agar, plates were placed in anaerobic chambers at 37 °C for 24 h, and CFU was determined. CFU results confirmed that significantly more growth of LP was observed when cells were grown in nutrient broth with 1:40 WH than in the presence of control ethanol (1.25%) (10^8^ vs. 10^5^ CFU/mL).

These results suggest that when nutrition is scarce, WH significantly promotes the growth of LP. However, when nutrition is plentiful, WH does not significantly affect the growth of the probiotic bacteria.

### 2.3. The Effect of whISOBAX (WH) on L. plantarum (LP) Biofilm Formation

Lactobacilli can form a biofilm in the gut microbiota, allowing them to persist during harsh environmental conditions and maintain their population density. To test for the effect of WH on biofilm formation by probiotic bacteria, LP was grown with or without WH in nutrient broth or in MRS. After 48 h, unbound cells were removed and adherent cells were stained. The OD was determined. Control cells were grown in broth alone or in various dilutions of 50% ethanol (0–2.5%). As shown in Figure 4a,b, WH significantly (*p* < 0.05) enhances biofilm formation in both MRS and nutrient broth. Plating of undiluted samples is shown in Figure 4c, indicating that significantly more bacteria grew in the presence of WH. See summary in Table 1.

### 2.4. The Effect of whISOBAX (WH) on L. plantarum (LP) Biofilm Formation on Alginate Beads

Calcium alginate hydrogels have been shown to improve viability of probiotic cells [23]. The effect of WH on biofilm formation on alginate beads was thus tested. These experiments were carried out by incubating bacteria in microtiter plates containing alginate beads, removing the beads into fresh plates, dissolving the beads, and evaluating bacterial optical density and number. An important consideration is that, to maintain bead integrity and stability during washes, these experiments could only be carried out with higher dilutions of WH. As shown in Figure 5, WH significantly (*p* < 0.05) enhanced LP biofilm on the beads.

### 2.5. The Effect WH on LP Growth in Aerobic and Anaerobic Environments

To compare the effect of WH on the growth of bacteria under aerobic vs. anaerobic conditions, bacteria were similarly grown in nutrient broth with WH or control ethanol solutions and microtiter plates placed in aerobic or anaerobic conditions. Anaerobic conditions were created using closed chambers containing anaerobic atmospheric generation bags. As shown in Figure 6, WH significantly (*p* < 0.01) enhanced bacterial growth in both aerobic and anaerobic conditions.

## 3. Discussion

We show here that the witch hazel extract whISOBAX^TM^ (WH) enhances the growth of the commercially available probiotic lactic acid bacteria *L. plantarum* strain Solace PS128 but suppresses the growth of pathogens such as *S. epidermidis* strain RP62A. In vitro, we show that WH significantly promotes the growth of *L. plantarum* in nutrient broth, which is a nutritionally poor environment for this bacterium. On the other hand, WH maintains the growth of *L. plantarum* when grown in MRS broth, which is a favorable, nutritionally rich environment. These results suggest that WH may help in the commercial processing of these bacteria and perhaps indicate that WH may help maintain gut bacterial homeostasis, or “normobiosis” [24].

*L. plantarum* does not form a biofilm on the tested abiotic surface (polystyrene), but in the presence of WH, biofilm formation was enhanced in a concentration-dependent manner, both in MRS and in Nutrient broth. Biofilms allow bacteria to persist under less favorable conditions as well. While a formation of a biofilm by pathogenic bacteria would be harmful, biofilm formation by probiotic bacteria would be considered beneficial to the maintenance of a healthy gut microbiome [25]. These results thus indicate that WH may increase the competitive advantage of the probiotic in the microbiome by supporting its ability to form a biofilm. Since formation of a biofilm allows bacteria to survive in the host under stressful conditions, WH gives this probiotic a competitive advantage to help it outlast pathogenic bacteria competing for nutrition.

Furthermore, we show here that WH enhances biofilm formation by LP on calcium alginate beads. Alginate is a commercially available anionic polysaccharide that is typically extracted from the cell walls of brown algae. It is a non-toxic, biocompatible, and biodegradable polymer and can be used as a probiotic delivery system because it is insoluble in acidic environments present in gastric juices but degrades slowly under conditions simulating the small intestinal fluid (pH 6.8) and rapidly under conditions simulating colonic fluid (pH 7.2) [26]. Alginate can thus provide protection for probiotics in extreme acidic environments while enabling the probiotic bacteria to establish themselves in the more favorable conditions of the intestinal tract [23,26,27].

WH enhances the growth of LP by yet unknown molecular mechanisms that can range from being a direct metabolic source, to scavenging oxygen, to selectively regulating gene expression; WH may also directly provide nutritional support to the bacteria, thus supporting growth under poor nutritional conditions. Alternatively, WH may protect LP from oxidative stress, because WH is rich in Hamamelitannin, which is a potent active oxygen scavenger [19,20,21,28]. Alternatively, WH may act as a signal transducer to a stress regulator, since hamamelitannin has been shown to inhibit the phosphorylation of TraP in staphylococci and its structural homologue YhgC/HmoB in bacilli, and both *traP* and *yhgC* genes were shown to be important for protecting bacteria from oxidative stress [7,8,9,10,29]. Perhaps regulating these pathways may result in selective advantages for survival in specific environmental niches, having antimicrobial properties to pathogenic bacteria such as *Staphylococci* while having prebiotic properties to probiotic bacteria such as *L. plantarum*.

Lactic acid bacteria are the most frequently used probiotics in fermented foods and beverages and as food supplements. During industrial processing and in the gastrointestinal tract, probiotic bacteria are exposed to potentially stressful environments, such as inadequate levels of oxygen, temperature, pH, or osmolarity and limited nutrition. These stressors affect their survival during processing and storage, as well as their ability to thrive in the gastrointestinal tract. To guarantee enough viable bacteria in the final product and effective health-promoting action in the host, it is necessary to either find strains that have high stress resistance or add a prebiotic or a growth enhancer that would enable them to selectively thrive. Although lactic acid bacteria are considered aerotolerant, their oxygen sensitivity is a major factor limiting their viability both during production and in the host. High oxygen levels can lead to the formation of reactive oxygen species (ROS), causing oxidative stress that results in damage to proteins, DNA, and lipids. To prevent oxidative stress, co-culturing with starter oxygen-depleting strains during fermentation has been used, but this approach has its own limitations, including the possible production of ROS species by these strains [30]. We show here that the effect of WH on the in vitro growth of LP was independent of the provided atmospheric conditions and its positive effect on bacterial growth was essentially the same for aerobic and anaerobic growth conditions. WH may thus enhance the growth of the probiotic bacteria during fermentation.

To summarize, probiotics are bacteria that, at adequate numbers, confer a health benefit to the host [13]. Prebiotics are defined as having “The selective stimulation of growth and/or activity(ies) of one or a limited number of microbial genus(era)/species in the gut microbiota that confer(s) health benefits to the host”. [11]. We show here that in vitro, WH inhibits the growth of pathogenic bacteria such as *S. epidermidis* strain RP62A but supports the growth of the probiotic *L. plantarum* strain Solace PS128, indicating that WH could potentially be used as a growth promoter of probiotic bacteria.

## 4. Materials and Methods

### 4.1. Bacteria

*S. epidermidis* ATCC 35984/RP62A was grown aerobically in Tryptic Soy Broth (TSB) with shaking at 37 °C to the early exponential phase of growth (OD_630_ 0.1), aliquoted, and frozen in −80 °C until use.

*L. plantarum* Solace PS128 (Oryx Biomedical, Fremont, CA, USA) was suspended in De Man Rogosa Sharpe (MRS) broth and plated on MRS agar plates. After a 24 h incubation at 37 °C, colonies were collected, resuspended in 15% Glycerol in Phosphate Buffered Saline (PBS) at OD_630_ 2.0, aliquoted, and frozen at −80 °C. Immediately before use, the bacteria were diluted 1:20 in nutrient broth.

Unless indicated otherwise, experiments (see below) were carried out in nutrient broth, which is a culture medium containing the basic nutrients 0.5% peptone and 0.3% beef extract, or in MRS broth, which is a rich culture broth containing 1% beef extract, 0.5% yeast extract, and 2% dextrose.

Bacteria were grown either under aerobic conditions in an atmospheric 37 °C incubator or in anaerobic conditions by first placing the microtiter plates in anaerobic chambers (using closed chambers containing anaerobic generation bags). Incubation times ranged from 1–3 days as indicated in results section.

### 4.2. Test Formulations

Test formulations were prepared from the following commercially available witch hazel extracts: WH (whISOBAX, StaphOff Biotech Inc., Hopkinton, MA, USA) containing 50 mg dry weight of witch hazel extract in 50% ethanol [19,20,21], CareOne (Salisbury, NC, USA) containing, as reported, witch hazel extract in 14% ethanol, and NFH (Nutritional Fundamentals for Health, Vaudreuil-Dorion, QC, Canada) containing, as reported, 60 mg dry weight in 20% ethanol. Test formulations were tested at final dilutions of 1:10 (CareOne) and 1:20–1:80 (WH, NFH and CareOne). Controls included culture broth supplemented with respective dilutions of 50% ethanol.

### 4.3. Bacterial Growth and Biofilm Formation

Growth and biofilm formation were tested as described [19,20,21]. Briefly, experiments were done in triplicates in polystyrene 96 -well plates (Costar, Corning Inc, Kennebunk, ME, USA) at a final volume of 200 µL. Bacteria were diluted in their respective culture broth to OD_630_ of 0.1 and tested at 25 µL/well. Culture broth was used as the blank instead of bacteria. Bacteria (or just culture broth) were incubated for the indicated times with increasing concentrations of witch hazel or its relevant solvent as a control. Growth was determined by optical density and confirmed by plating.

To test for biofilm, unbound bacteria were removed, wells washed three times in PBS, and adherent cells fixed in methanol, and stained in crystal violet. Stained cells were vigorously washed with water, dried in air, and degraded in 0.1% SDS to release the stain, and color intensity was determined spectrophotometrically at OD_630_.

For quality assessment, 1 µL of undiluted bacterial samples were streaked on MRS agar plates and grown for a day in an anaerobic chamber at 37 °C. To determine the number of colony-forming units (CFU), bacterial samples were serially diluted in peptone water, 100 µL of diluted cultures were plated on MRS agar plates, and plates were incubated in an anaerobic chamber at 37 °C for 48 h.

### 4.4. Preparation and Analysis of Biofilm on Alginate Beads

#### 4.4.1. Bead Preparation

Sterile solutions of 4% Sodium Alginate in water and 0.1 M CaCl_2_ were prepared. CaCl_2_ solution was placed in polystyrene microtiter 96-well plates (150 µL per well). To produce a single bead per well, a drop of the alginate solution was placed in each well using a 3cc syringe with a 22-gauge needle. Plates were kept for 2–3 days at 4 °C before use.

#### 4.4.2. Biofilm Formation on Alginate Beads

To prepare the beads for testing, CaCl_2_ solution was carefully removed, beads were washed with cold sterile water (200 µL × 2), and test solutions were then added as indicated, with a final volume of 200 µL per well containing a single bead. All experiments were carried out in triplicates.

#### 4.4.3. Biofilm Analysis of Beads

Once bacteria were incubated with the beads, unbound solution was removed from the wells, the beads were washed twice with saline, and finally, the beads were removed to a fresh plate using the HBTD method (see below) to allow for specific analysis of the biofilm on the bead itself.

#### 4.4.4. HBTD (Howard Balaban Tape and Dump) Method

A fresh sterile plate was placed on top of the bead containing plate, making sure that wells were perfectly aligned. The plates were then securely taped to one another, being careful to maintain the alignment of the wells. Once secured, gently, but swiftly, the plates were rotated so the original plate that had contained the beads was now on top. Next, if beads were still left in the original plate, the wells were carefully tapped to cause the beads to fall to the mirror plate. Once all the beads were transferred, the tape was removed and the fresh mirror plate further analyzed as follows: beads were dissolved by adding 100 µL of sterile 1% citric acid for 20 min at room temperature and serially diluted in broth, and CFU was determined by plating.

### 4.5. Statistical Analysis

All experiments were carried out at least three times, and each condition was tested in triplicate. Averages are presented. Standard deviation values were calculated using the Microsoft Excel “*n* − 1” method, and significance was calculated using Microsoft Excel two-tailed Student’s t-test, where *p* < 0.05 was considered as a significant difference.

## Figures and Tables

**Figure 1 antibiotics-11-00395-f001:**
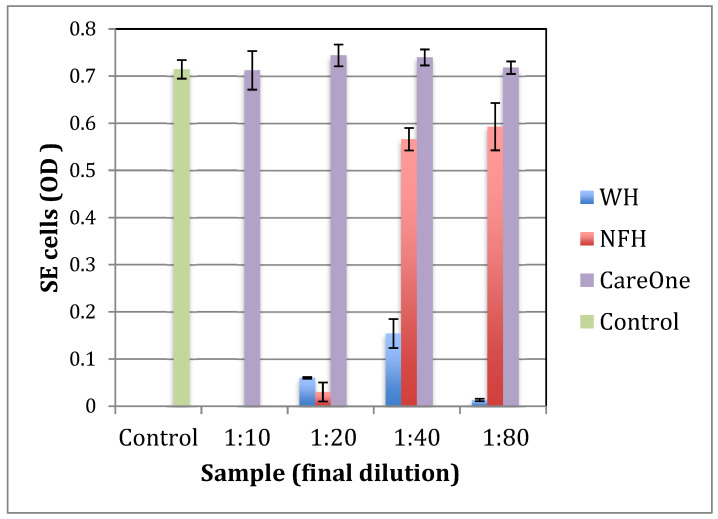
The effect of witch hazel extracts on the growth of *S. epidermidis.* Bacteria were grown overnight in TSB in the presence of increasing amounts of witch hazel extracts, ethanol, or culture broth alone (control). OD_630_ was determined after 24 h. Witch hazel extracts tested: WH (whISOBAX, StaphOff Biotech), CareOne extract (CareOne), and NFH (Nutritional Fundamentals for Health).

**Figure 2 antibiotics-11-00395-f002:**
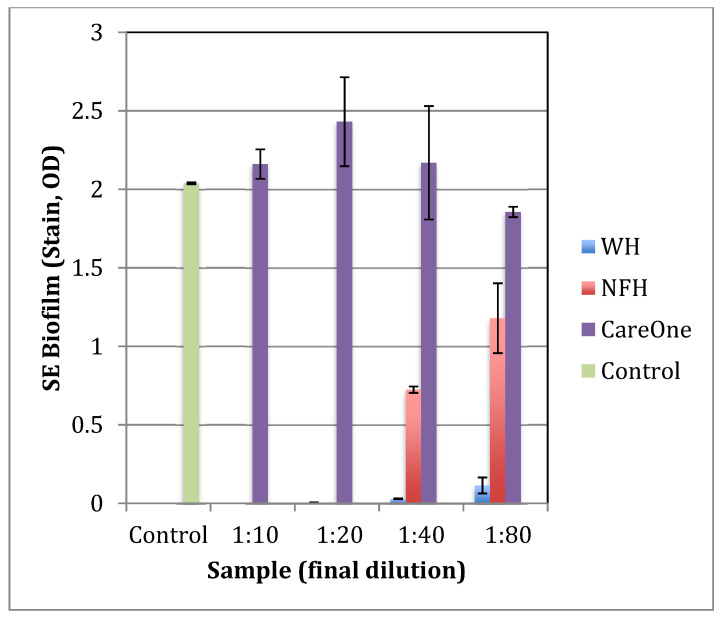
The effect of witch hazel extracts on biofilm formation by *S. epidermidis*. Bacteria were grown overnight in TSB in the presence of increasing amounts of witch hazel extracts, ethanol, or culture broth alone (control). After 24 h, unbound bacteria were removed, biofilm bacteria stained in crystal violet, stain dissolved in SDS and OD_630_ were determined. Witch hazel extracts tested: WH (whISOBAX, StaphOff Biotech), CareOne extract (CareOne), and NFH (Nutritional Fundamentals for Health).

**Figure 3 antibiotics-11-00395-f003:**
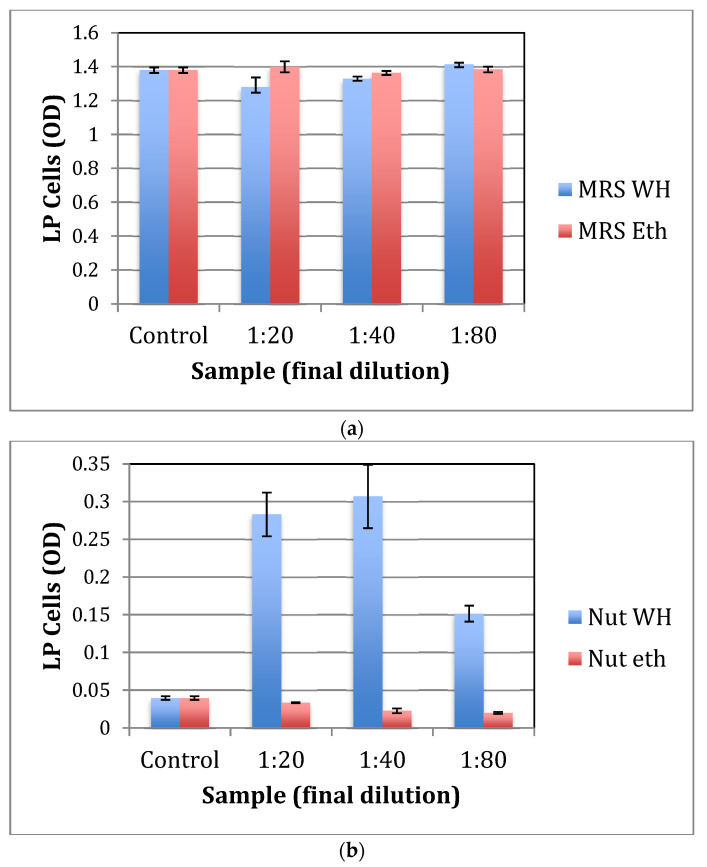
The effect of witch hazel on the growth of *L. plantarum*. (**a**) Bacteria were grown overnight in MRS both in the presence of increasing amounts of whISOBAX, ethanol (50%, starting solution) or culture broth alone (control). OD_630_ was determined after 48 h. (**b**) Bacteria were grown overnight in nutrient both (Nut) in the presence of increasing amounts of whISOBAX, ethanol (50%, starting solution) or culture broth alone (control). OD_630_ was determined after 48 h.

**Figure 4 antibiotics-11-00395-f004:**
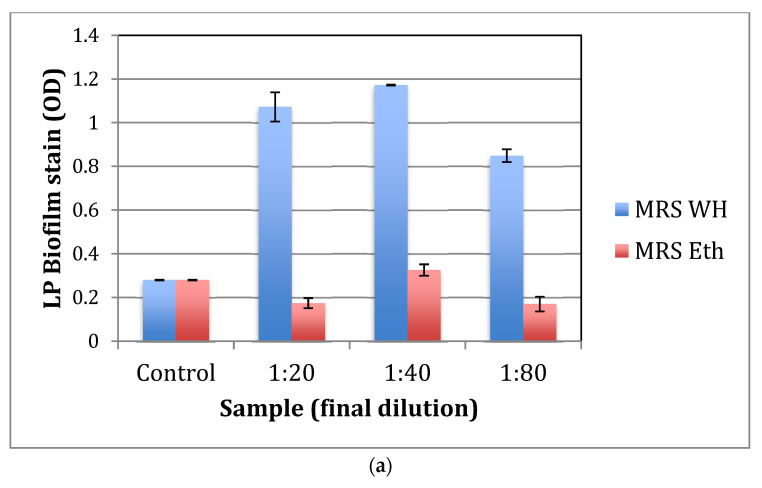
The effect of witch hazel on biofilm formation by *L*. *plantarum.* Bacteria were grown overnight in MRS (**a**) or nutrient broth (Nut) (**b**) in the presence of increasing amounts of whISOBAX (WH), ethanol (Eth), or medium alone (control). After 48 h, unbound bacteria were removed, bound (biofilm) bacteria were stained in crystal violet, stain was dissolved in SDS, and OD was determined. (**c**) Bacteria were grown with a final 1:20 dilution of WH or Eth control (see (**b**) above). Bound cells were resuspended in nutrient broth, and undiluted samples were streaked on MRS agar plates, which were placed in anaerobic chambers for 24 h.

**Figure 5 antibiotics-11-00395-f005:**
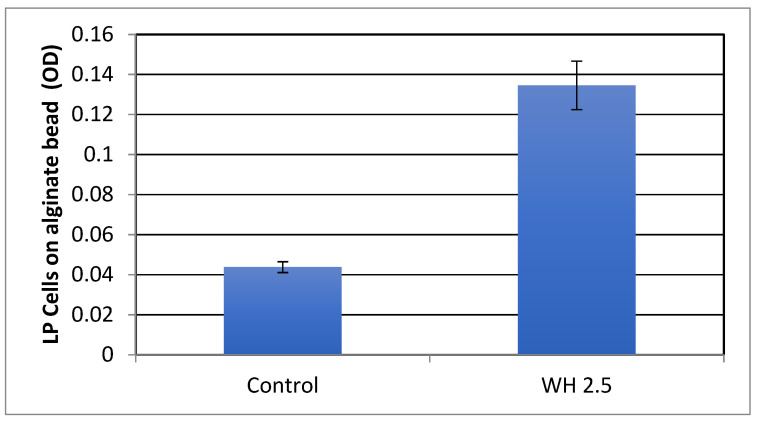
The effect of witch hazel (WH) on biofilm formation by *L*. *plantarum* on alginate beads. Bacteria were incubated with alginate beads in the presence of WH diluted 1:80 (WH 2.5) or ethanol control and the OD of adherent bacteria was determined.

**Figure 6 antibiotics-11-00395-f006:**
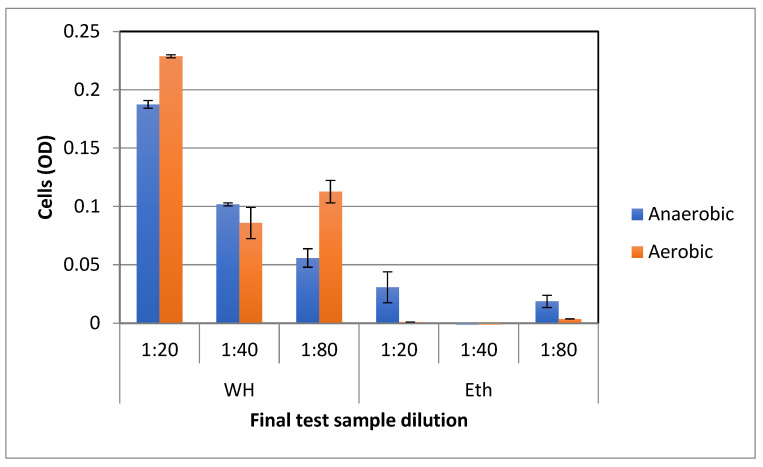
The effect of witch hazel on anaerobic vs. aerobic growth of *L*. *plantarum* in nutrient broth. Bacteria were grown overnight in nutrient broth in the presence of increasing amounts of whISOBAX (WH) or ethanol (Eth) control. After 48 h, OD was determined.

**Table 1 antibiotics-11-00395-t001:** Summary of the approximate ratio of *L. plantarum* cells grown in the presence of whISOBAX (WH) or ethanol (control) in each broth tested (nutrient broth (Nut) or MRS broth).

	MRS Broth	Nutrient Broth
	Control:WH	Control:WH
Growth	1:1	1:6
Biofilm	1:4	1:10

## Data Availability

Data is available upon request.

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
