# Peer review of "The Effect of Tannin-Rich Witch Hazel on Growth of Probiotic Lactobacillus plantarum"

_antibiotics, 2022, doi:10.3390/antibiotics11030395_

Round 1

Reviewer 1 Report

The authors have performed some experiments to improve the manuscript. The study was improved, however some of the conclusions are not supported by these investigations. The authors conclude WH as a prebiotic while using a pathogen which is generally associated with skin microbiota, whereas gut pathogens might be a better candidate for proving such significance. In addition, the criteria for a prebiotic candidate needs more rigorous investigations and the impact should be demonstrated in host.

The authors have addressed most of the comments, however still some modifications are necessary before paper may be accepted.

The title of study can be changed, citing clearly the L. plantarum strains.

Strain name for L. plantarum and Staphylococcus epidermis should be mentioned in the manuscript Methods and Discussion.

The strain name for L. plantarum may also mentioned in the Title.

Its potential as prebiotic needs further investigations, therefore authors may discuss it in discussions if they want.

L125. Reference should be removed from the middle of the phrases, here and elsewhere as well.

L126-129, ambiguous

L199 Reference should be removed from the middle of the phrases, here and elsewhere as well.

L349. Lactic acid bacteria

The study design yet indicates the significance of WH as a growth supplement for a commercial probiotic strain. These findings are of interest to the scientific community. Therefore, the conclusions drawn from this study need to modified. The word prebiotic should not be used for WH, however a supplement or growth enhancer might be referred.

L468. WH may be used as prebiotic should be rephrased appropriately, since it does not qualify to be a prebiotic. The criterion to be labelled as prebiotic needs further investigations. However, it may be modified as a supplement for growth of a probiotic L. plantarum.

Author Response

Dear Editor,

Thank you very much for the reviewers’ comments. Below are their comments along with our responses.

Sincerely,

Dr. Reuven Rasooly

Reviewer 1
- The authors have performed some experiments to improve the manuscript. The study was improved, however some of the conclusions are not supported by these investigations. The authors conclude WH as a prebiotic while using a pathogen which is generally associated with skin microbiota, whereas gut pathogens might be a better candidate for proving such significance. In addition, the criteria for a prebiotic candidate needs more rigorous investigations and the impact should be demonstrated in host.

RESPONSE: the manuscript has been edited to read: “ … indicating that WH can potentially be used as a growth promoter of probiotic bacteria. “

- The title of study can be changed, citing clearly the L. plantarum strains.

RESPONSE: this information was added to the abstract.

- Strain name for L. plantarum and Staphylococcus epidermis should be mentioned in the manuscript Methods and Discussion.

RESPONSE: strain names have been added.

- L125. Reference should be removed from the middle of the phrases, here and elsewhere as well.

CORRECTED

- L126-129, ambiguous

CORRECTED

- L199 Reference should be removed from the middle of the phrases, here and elsewhere as well.

CORRECTED

- L349. Lactic acid bacteria

CORRECTED

-The study design yet indicates the significance of WH as a growth supplement for a commercial probiotic strain. These findings are of interest to the scientific community. Therefore, the conclusions drawn from this study need to modified. The word prebiotic should not be used for WH, however a supplement or growth enhancer might be referred.

CORRECTED

Reviewer 2 Report

The manuscript has been revised according to the suggestions and should be published.

Author Response

Thank you for reviewer

Reviewer 3 Report

In this new version of the manuscript submitted to the Special Issue "10th Anniversary of Antibiotics - Recent Advances in Novel Antimicrobial Agents" the authors included important information, such as the concentration of Hamamelitannin in WH and NFH extracts, but did not test the activity of fractions of these extracts or of Hamamelitannin isolated (since this compound is commercialized). The presentation of the results of these tests with fractions and especially with Hamamelitannin would help to elucidate the unknown molecular mechanisms cited in the discussion (lines 431-443) and would be more engaged in the purpose of this special issue as "Novel Antimicrobial Agents".
Furthermore, all activities reported against foodborne pathogenic bacteria ( S. epidermidis, E. coli and K. pneumoniae) have already been reported by other authors. And, the potential use as a prebiotic could be shown with other bacteria that compose the intestinal microbiota besides L. plantarum, could have evaluated molecular mechanisms envolved, as well as could have used basic toxicity tests of Hamamelitannin.

Author Response

Reviewer 3

- The authors included important information, such as the concentration of Hamamelitannin in WH and NFH extracts, but did not test the activity of fractions of these extracts or of Hamamelitannin isolated (since this compound is commercialized). The presentation of the results of these tests with fractions and especially with Hamamelitannin would help to elucidate the unknown molecular mechanisms cited in the discussion (lines 431-443) and would be more engaged in the purpose of this special issue as "Novel Antimicrobial Agents". Furthermore, all activities reported against foodborne pathogenic bacteria ( S. epidermidis, E. coli and K. pneumoniae) have already been reported by other authors. And, the potential use as a prebiotic could be shown with other bacteria that compose the intestinal microbiota besides L. plantarum, could have evaluated molecular mechanisms envolved, as well as could have used basic toxicity tests of Hamamelitannin.

RESPONSE: Thank you for reviewers’ suggestions. Unfortunately, these studies are beyond the scope of this manuscript.

Reviewer 4 Report

I can find that most of the recommendations were implemented beside  recommendations from other reviewers but there are two comments:

1) The lines from 166-204 should be transfered to suitable sections in the methodology part

2)In my previous revision , I mentioned that biofilm should be correlated to growth by that I mean using biofilm index for expressing biofilm values as in this reference "Yaikhan, T.; Chuerboon, M.; Tippayatham, N.; Atimuttikul, N.; Nuidate, T.; Yingkajorn, M.; Tun, A.W.; Buncherd, H.; Tansila, N.
Indole and Derivatives Modulate Biofilm Formation and Antibiotic Tolerance of Klebsiella pneumoniae. Indian J. Microbiol. 2019,
59, 460–467."

Author Response

Reviewer 4

1) The lines from 166-204 should be transferred to suitable sections in the methodology part

CORRECTED

2)In my previous revision , I mentioned that biofilm should be correlated to growth by that I mean using biofilm index for expressing biofilm values as in this reference "Yaikhan, T.; et al., .Indole and Derivatives Modulate Biofilm Formation and Antibiotic Tolerance of Klebsiella pneumoniae. Indian J. Microbiol. 2019, 59, 460–467."

RESPONSE: We chose here to present biofilm intensity using conventional methods of a single OD read instead of an interesting idea presented in your suggested article using index biofilm of OD595/600.

Reviewer 5 Report

Congratulations to the authors for this interesting work with the aim of using WH as a prebiotic. Here are my comments:

-Page 1 and 2: why this jump occurs, appearing only the names and the name of the authors on page 1; while on page 2 their affiliations and abstract appear. Please put it all on the same page.
-Keywords: please put in alphabetical order. It could also be interesting to add the genus Hamamelis to the keywords.
-Lines 40-41 add a reference.
-Line 50 to 54 have another line spacing
-Line 64: change have to has
-Line 65: put the references [11,12,13] as follows [11-13]
-Lines 78 to 82 have another line spacing
-Line 121 to 124: put in the materials and methods section, not in the results section.
-Figure 3a and 3b: I suggest putting a caption on each figure (3a and 3b) so that the figure is more intuitive, even if it is repetitive.
-Table 2: Please follow the mpdi-Antibiotics guidelines for formatting the table (https://www.mdpi.com/journal/antibiotics/instructions). In addition, the text of table 1 must be above the table, not below (as it happens with figures). It would also be advisable to add the meaning of Nut and WH below the table.
-Figure 4, do the same as with Figure 3. Put a caption on each figure to make the explanation more intuitive. Does figure 3c correspond to what concentration of WH?
-Line 173: the indentation is missing.
-Figure 5: What does the 2.5 of WH mean?
-Line 192: put the p-value in italics
-Figure 6: the one that corresponds to Eth at 1:40, is it negative or is it zero? Also add in the foot of the figure the meaning of WH, Eth and their corresponding proportions.
-Line 197: remove the italics in 'in nutrient broth'
-Line 200 to 208: justify the paragraph and remove the bold.
-Line 202: put in vitro in italics
-Line 236: put the references [7,8,9] as [7-9]
-Lines 263 to 268: conclude (point 5) that is missing in the manuscript and expand it.
-Line 278: please explain in more detail the growing conditions: time, temperature,…
-Line 295: you can indicate in detail the incubation times that were used in protocol 4.3
-Line 342: put the p-value in italics
-Section 4.4. put each subsection as:
        4.4.1. Bead preparation
        4.4.2. Biofilm formation on alginate beads
        4.4.3. Biofilm analysis of beads
        4.4.4. HBTD (Howard Balaban Tape and Dump)
-Line 338: why the statistical study is used using Excel and not another more specific statistical program such as SPSS, XLStat.
-References: please put all of them in the same format following the guidelines of mdpi-antibiotics since there are some that put the date, in all the links the link is not hyperlinked, the line spacing as well as the space between them is not homogeneous. Change reference 4 to a bibliographical reference like this one. Also, there are some references that are quite old, I suggest to update some of them like:
        Reference 4: doi: 10.1007/s10096-019-03539-6
        Reference 6: doi: 10.1128/CMR.00134-14
        Reference 7: doi: 10.1002/iid3.456
        References 15-17: this reference can also be added on the metabolism of gut microbiata: doi: 10.1002/jsfa.10378

Author Response

Reviewer 5
-Page 1 and 2: why this jump occurs, appearing only the names and the name of the authors on page 1; while on page 2 their affiliations and abstract appear. Please put it all on the same page.

CORRECTED

-Keywords: please put in alphabetical order. It could also be interesting to add the genus Hamamelis to the keywords

CORRECTED

-Lines 40-41 add a reference.

CORRECTED

-Line 50 to 54 have another line spacing

CORRECTED

-Line 64: change have to has

CORRECTED

-Line 65: put the references [11,12,13] as follows [11-13]

CORRECTED

-Lines 78 to 82 have another line spacing

CORRECTED

-Line 121 to 124: put in the materials and methods section, not in the results section.

CORRECTED
-Figure 3a and 3b: I suggest putting a caption on each figure (3a and 3b) so that the figure is more intuitive, even if it is repetitive.

CORRECTED

-Table 2: Please follow the mpdi-Antibiotics guidelines for formatting the table (https://www.mdpi.com/journal/antibiotics/instructions). In addition, the text of table 1 must be above the table, not below (as it happens with figures). It would also be advisable to add the meaning of Nut and WH below the table.

CORRECTED

-Figure 4, do the same as with Figure 3. Put a caption on each figure to make the explanation more intuitive. Does figure 3c correspond to what concentration of WH?

CORRECTED

-Line 173: the indentation is missing.

CORRECTED

-Figure 5: What does the 2.5 of WH mean?

CORRECTED

-Line 192: put the p-value in italics

CORRECTED

-Figure 6: the one that corresponds to Eth at 1:40, is it negative or is it zero? Also add in the foot of the figure the meaning of WH, Eth and their corresponding proportions.

CORRECTED

-Line 197: remove the italics in 'in nutrient broth'

CORRECTED

-Line 200 to 208: justify the paragraph and remove the bold.

CORRECTED

-Line 202: put in vitro in italics

CORRECTED

-Line 236: put the references [7,8,9] as [7-9]

CORRECTED

-Lines 263 to 268: conclude (point 5) that is missing in the manuscript and expand it.

CORRECTED
-Line 278: please explain in more detail the growing conditions: time, temperature,…

CORRECTED

-Line 295: you can indicate in detail the incubation times that were used in protocol 4.3

RESPONSE: ADDED DETAILS IN SECTION 4.1

-Line 342: put the p-value in italics

CORRECTED

-Section 4.4. put each subsection as:4.4.1 etc

CORRECTED

-Line 338: why the statistical study is used using Excel and not another more specific statistical program such as SPSS, XLStat. While we realize that xx can provide more sophisticated tools,

RESPONSE: As the differences between experimental and controls were visibly significant, we chose to use statistics tools that, in this case, are as effective in Excel.

-References: please put all of them in the same format following the guidelines of mdpi-antibiotics since there are some that put the date, in all the links the link is not hyperlinked, the line spacing as well as the space between them is not homogeneous.

RESPONSE: References were updated as appropriate, and the reference formatting has been corrected to match template and format guidelines. Hyperlinks have been removed.

Round 2

Reviewer 1 Report

Prebiotic has been deleted from conclusions but should be deleted from the abstract as well. I still find it in abstract.

'These results indicate that WH may thus be used as a prebiotic to enhance the growth of probiotics'.

The strain name for L. plantarum has not been provided in abstract.

Author Response

Dear Editor,

Thank you very much for the reviewers’ comments. Both corrections made (see attached revised article): Prebiotic mention removed also in abstract and strain name added in Abstract.

Sincerely,

Dr. Reuven Rasooly

Reviewer 5 Report

Thank you very much for sending me the corrections of the article. The authors have followed the suggestions sent, increasing the quality and organization of the article. Therefore, it can be accepted in its present form.

Author Response

Thank you very much